# Performance of 11 Host Biomarkers Alone or in Combination in the Diagnosis of Late-Onset Sepsis in Hospitalized Neonates: The Prospective EMERAUDE Study

**DOI:** 10.3390/biomedicines11061703

**Published:** 2023-06-13

**Authors:** Sylvie Pons, Sophie Trouillet-Assant, Fabien Subtil, Fatima Abbas-Chorfa, Elise Cornaton, Amélie Berthiot, Sonia Galletti, Aurélie Plat, Stephanie Rapin, Laurene Trapes, Laurence Generenaz, Karen Brengel-Pesce, Arnaud Callies, Franck Plaisant, Olivier Claris, Aurelie Portefaix, Cyril Flamant, Marine Butin

**Affiliations:** 1Joint Research Unit Hospices Civils de Lyon-bioMérieux, 69795 Pierre Bénite, France; sylvie.pons@biomerieux.com (S.P.); sophie.trouillet-assant@chu-lyon.fr (S.T.-A.); laurence.generenaz@biomerieux.com (L.G.); karen.brengel-pesce@biomerieux.com (K.B.-P.); 2Centre International de Recherche en Infectiologie (CIRI), Université Claude Bernard Lyon 1, INSERM U1111, CNRS UMR5308, ENS Lyon, 69364 Lyon, France; 3Service de Biostatistique, Hospices Civils de Lyon, 69003 Lyon, France; fabien.subtil@chu-lyon.fr (F.S.); fatima.abbas@chu-lyon.fr (F.A.-C.); 4Laboratoire de Biométrie et Biologie Évolutive, CNRS UMR 5558, Université Claude Bernard Lyon 1, 69622 Villeurbanne, France; 5Department of Neonatology, Hospices Civils de Lyon, Hôpital Femme Mère Enfant, 69677 Bron, France; elise.cornaton@chu-lyon.fr (E.C.); franck.plaisant@chu-lyon.fr (F.P.); 6Clinical Investigation Center CIC 1407, Université de Lyon and Hospices Civils de Lyon, 1407 Inserm, UMR 5558, LBBE, CNRS Lyon, 69677 Bron, France; amelie.berthiot@chu-lyon.fr (A.B.); sonia.galletti@chu-lyon.fr (S.G.); aurelie.portefaix@chu-lyon.fr (A.P.); 7Department of Neonatology, University Hospital of Saint Etienne, 42055 Saint Etienne, France; aurelie.cantais@chu-st-etienne.fr (A.P.); stephanie.rapin@chu-st-etienne.fr (S.R.); laurene.trapes@chu-st-etienne.fr (L.T.); 8Department of Neonatology, Hôpital Mère-Enfant, University Hospital of Nantes, 44093 Nantes, France; arnaud.callies@chu-nantes.fr (A.C.); cyril.flamant@chu-nantes.fr (C.F.); 9Department of Neonatology, Hospices Civils de Lyon, Hôpital Croix Rousse, 69002 Lyon, France; olivier.claris@chu-lyon.fr; 10Research Unit EA 4129, University Claude Bernard Lyon 1, 69622 Villeurbanne, France

**Keywords:** late-onset sepsis, LOS, neonatal intensive care unit, NICU, biomarker

## Abstract

Despite the high prevalence of late-onset sepsis (LOS) in neonatal intensive care units, a reliable diagnosis remains difficult. This prospective, multicenter cohort study aimed to identify biomarkers early to rule out the diagnosis of LOS in 230 neonates ≥7 days of life with signs of suspected LOS. Blood levels of eleven protein biomarkers (PCT, IL-10, IL-6, NGAL, IP-10, PTX3, CD14, LBP, IL-27, gelsolin, and calprotectin) were measured. Patients received standard of care blinded to biomarker results, and an independent adjudication committee blinded to biomarker results assigned each patient to either infected, not infected, or unclassified groups. Performances of biomarkers were assessed considering a sensitivity of at least 0.898. The adjudication committee classified 22% of patients as infected and all of these received antibiotics. A total of 27% of the not infected group also received antibiotics. The best biomarkers alone were IL-6, IL-10, and NGAL with an area under the curve (95% confidence interval) of 0.864 (0.798–0.929), 0.845 (0.777–0.914), and 0.829 (0.760–0.898), respectively. The best combinations of up to four biomarkers were PCT/IL-10, PTX3/NGAL, and PTX3/NGAL/gelsolin. The best models of biomarkers could have identified not infected patients early on and avoided up to 64% of unjustified antibiotics. At the onset of clinical suspicion of LOS, additional biomarkers could help the clinician in identifying non-infected patients.

## 1. Introduction

Late-onset sepsis (LOS) is frequent in neonatal intensive care units (NICUs), especially in the most preterm and lowest birth weight infants, and can lead to life-threatening issues [1]. The diagnosis of LOS at onset is a challenge since it relies mainly on clinical signs that are neither specific nor constant, including respiratory distress, temperature instability, as well as neurological or hemodynamic disorders [2]. Moreover, in this population, it is difficult to differentiate signs of infection from clinical signs related to other medical conditions, especially in very low birth weight (<1500 g) infants. Blood culture is considered the gold standard for the diagnosis of LOS; however, the time to result is long (up to 48 h), in line with the time needed for culture [3]. In this context, and in the absence of a test with high negative predictive value (NPV) providing immediate results, antibiotics are frequently administered to neonates suspected of having LOS before the result of the blood culture is available in order to avoid a rapid clinical deterioration [4]. This leads to unnecessary exposure to antibiotics; for example, it was reported in a Canadian NICU that 85% of very low birth weight infants were exposed to antibiotics during their hospitalization, among whom 75% were not infected [5]. This is worrying given the negative impact of even short antibiotic exposure at the early stage of life on the gut microbiota at the time of its implementation and the associated risk of developing asthma, allergic diseases, and metabolic disorders [6,7].

The use of biomarkers could help clinicians recognize true infections in neonates and thus decrease the prescription of unjustified antibiotics. Several studies have been published concerning the value of biomarkers in neonatal sepsis [8]. In particular, C-reactive protein (CRP) has been widely used for many years but has a poor performance for the diagnosis of LOS at the onset of clinical signs, probably because of both the delay between the onset of sepsis and the rise of CRP level, as well as numerous other situations in which CRP increases [9]. This is illustrated by a recent meta-analysis that found that in a hypothetical cohort of 1000 neonates, assessing serum CRP level alone would miss 152 cases of infection (false-negative result) and wrongly diagnose 156 cases (false-positive result) [10]. Furthermore, studies investigating biomarkers evaluated the performance of these in the early diagnosis of LOS, but not the ability of biomarker-based protocols to rule out the diagnosis of LOS [4]. To avoid the prescription of antibiotics in non-infected patients, a biomarker with excellent sensitivity and negative predictive value is therefore needed. The primary objective of the present study was to identify the best combination of biomarkers or single biomarkers, among 11 host biomarkers selected based on their reported performance in the literature, that can rule out the diagnosis of LOS in hospitalized neonates with a clinical suspicion of LOS early on.

## 2. Materials and Methods

### 2.1. Study Design and Participants

A prospective, multicenter cohort study, named EMERAUDE (Evaluation of bioMarkErs to Reduce Antibiotics Use in hospitalizeD nEonates), was conducted in two French NICUs (Hôpital Femme Mère Enfant, Bron, France; Hospices Civils de Lyon, Lyon, France; Centre Hospitalier Universitaire de Nantes, Nantes, France) between 19 November 2017 and 20 November 2020 (ClinicalTrials.gov ID: NCT03299751). Written informed consent was obtained from at least one of the parents or legal guardians. The study was approved by a French ethics committee (Comité de Protection des Personnes [CPP Sud-Ouest et Outremer III]) under the registration number 2017-A02492-51 and was conducted according to the recommendations of Good Clinical Practice and the Declaration of Helsinki.

Hospitalized neonates of ≥7 days of life with suggestive signs of LOS requiring a blood culture were consecutively included. The decision to sample a blood culture because of a suspected LOS was at the discretion of the attending neonatologist and was usually based on the following signs: fever > 38 °C, tachycardia > 160 beats per minute, capillary refill time >3 s, gray and/or pale skin complexion, apnea or bradycardia events, abdominal bloating, rectal bleeding, hypotonia or lethargy, seizures without other obvious cause, increased respiratory support and/or increased FiO2, cutaneous rash, inflammation at the needle-puncture site of the central venous catheter. Of note in these NICUs is the volume of blood recommended for a blood culture, which is at least 1 mL per bottle. A consent form signed by at least one parent/legal representative was also mandatory to include the patient. Exclusion criteria were treatment with antibiotics for a bacteriologically confirmed infection during the previous 48 h prior to inclusion, as well as surgery or vaccination during the 7 days prior to inclusion. Patients with invalid inclusion criteria were excluded from the study, as well as those without analyzable blood samples.

The characteristics of patients at the time of inclusion and between 48 and 72 h were collected, including demographics, medical history, disease history, physical examination, and results of the blood culture. Results of other tests that could have been performed for routine care (chest X-ray, bacteriological samples, CRP, white blood cell count, absolute neutrophil count) were also collected, as was the decision whether or not to treat the patient with antibiotics, which was at the discretion of the physician on the basis of medical history of the patient, clinical characteristics, and CRP level. Of note, vancomycin is usually recommended as the first line antibiotic in the study’s NICU patients given the microbial epidemiology, showing a majority of coagulase negative staphylococci in LOS.

### 2.2. Sample Collection and Biomarker Measurement

For each included patient, at the time of the venipuncture prescribed for standard care, up to 0.4 mL of blood was collected in BD™ Microtainer™ Serum Separating Tubes (Becton Dickinson, Franklin Lakes, NJ, USA; reference BD365968). After 2 h clotting at room temperature and centrifugation at 2500× *g* for 10 min, sera were aliquoted and stored frozen at −80 °C until the measurement of 11 biomarkers. The concentrations of procalcitonin (PCT), interferon gamma inducible protein 10 (IP-10), interleukin 6 (IL-6), interleukin 10 (IL-10), neutrophil gelatinase-associated lipocalin (NGAL), pentraxin 3 (PTX3), presepsin (CD14), and lipopolysaccharide-binding protein (LBP) were measured in serum using customized multiplexed assays in the ELLA Automated Immunoassay System with the Simple Plex Technology (Protein Simple, San Jose, CA, USA), according to the manufacturer’s instructions. ELLA platform is an integrated immunoassay system that consists of a disposable microfluidic cartridge for biomarker assays for either single- or multi-analyte quantitation and an automated analyzer, the ELLA instrument. PCT, IP-10, IL-6, and IL-10 quantitation was simultaneously performed in a multiplex cartridge format using 50 µL of two-fold diluted serum. NGAL, PTX3, CD14, and LBP concentrations were measured in a multiplex cartridge format using 50 µL of 1:400 diluted serum. Gelsolin levels were measured by using the Human GS (Gelsolin) ELISA kit (Elabsciences, Houston, TX, USA) using 1:2000 diluted serum. Calprotectin concentrations were measured using the Human S100A8/S100A9 Heterodimer Quantikine ELISA kit (R&D Systems, Minneapolis, MN, USA) with a 1:200 diluted serum. IL-27 levels were measured by using the DuoSet ELISA kit (R&D Systems, Mineapolis, MN, USA) using a two-fold-diluted serum. All measurements were performed according to the manufacturer’s instructions and in duplicate per conventional ELISA. The selection of these biomarkers was based on the results of previous studies about their value in the context, as well as the absence of variation related to gestational or postnatal age, and an increase in cases of infectious disease [11,12,13,14,15,16,17,18,19,20,21].

### 2.3. Definitions

The primary outcome was the diagnosis of LOS determined by an adjudication committee composed of three neonatologist experts, independent of the management of neonates in the study centers. A positive blood culture was considered insufficient to confirm LOS because of the risk of contaminant, especially for coagulase negative staphylococci. Moreover, there is no consensual definition of LOS based on clinical signs and/or biomarkers results. Therefore, in the present study, the classification of patients as infected, not infected, or unclassified was performed by the adjudication committee based on the clinical and microbiological data, as well as on the CRP level in serum collected at inclusion and after 48 h, blinded to the values of the study biomarkers and to the decision of their peers. Final diagnosis depended on each of the three classifications following a predefined process, as detailed in Appendix B. The diagnostic performance of the biomarker combinations and of the clinical signs were based on the classification by the adjudication committee.

### 2.4. Statistical Analysis

Continuous variables were described by the median and range, and qualitative variables by count and percentage. Comparisons between groups were made using the Kruskal–Wallis or Wilcoxon tests for continuous variables and Chi-squared or Fisher’s exact test for qualitative variables. The diagnostic accuracy of evaluated biomarkers, of clinical signs, and of CRP (as part of standard care) was assessed in the groups of infected and not infected patients. Univariate logistic regression was used to assess the association between clinical signs and confirmed infection; the association was quantified by odds ratio (OR) with 95% confidence intervals [95% CI]. Clinical signs with a *p*-value < 0.20 with low collinearity were included in a multivariate model. Biomarkers were combined through logistic regression models to predict the infection status, considering an additive effect on the logistic scale. Logarithmic transformations were applied when necessary to fulfill the hypotheses of the model. Predictions of the model (predicted probabilities of infection) were then used as a new marker. Receiver operating characteristic curves (ROC) were built to estimate the performance of the clinical signs, CRP, biomarkers of the study, or combination of biomarkers for the diagnosis of infection. The area under the curve (AUC) and partial AUC (part of the curve for which the sensitivity is ≥0.898) were then calculated [22]. For each biomarker of the study (or combination of biomarkers), the threshold with the highest specificity and a sensitivity ≥0.898 was estimated (for combination of biomarkers, the threshold of predicted infection probability) with the associated specificity, positive and negative predictive value, and positive and negative likelihood ratios. A cut-off of at least 0.898 was defined for the sensitivity in order to identify the best biomarker alone or in combination to rule out the diagnosis of LOS in symptomatic neonates early on. The optimism, the fact that the model gives better predictions on the data used to build the model than on independent datasets, was assessed by 20-times 5-fold cross validation [23]. Statistical analyses were performed using R software, version 4.0.2 (R Foundation for Statistical Computing, Vienna, Austria) and SAS Institute software, version 9.4 (Cary, CN, USA). A heatmap was generated by scaling and centering log10-transformed biomarker concentrations, and the dendogram was drawn based on hierarchical clustering analysis (Euclidean distance matrix with Ward’s method) using Partek^®^ Genomics Suite^®^ software version 7.0 (Partek Inc., St. Louis, MO, USA).

## 3. Results

### 3.1. Patients’ Characteristics

A total of 234 hospitalized neonates with suspected LOS were included, of which 230 had analyzable samples (Figure 1). They were mainly preterm (80%) with a median (range) gestational age of 27 (23–41) weeks and a median (range) birth weight of 940 (450–4660) g Out of the 230 analyzed patients, 137 (59.6%) were boys (Table 1). Suspicion of LOS occurred at a median (range) of 14 (7–178) days of age. The most frequent signs related to suspicion of LOS were tachycardia (124/230, 53.9%), bloating/rectal bleeding (120/230, 52.2%), apnea or bradycardia events (111/229, 48.5%), and increased respiratory support and/or FiO2 (107/230, 46.5%).

### 3.2. Demographics and Microbiological Characteristics According to Infection Status

The adjudication committee classified 51 (22.2%) neonates as infected, 153 (66.5%) as not infected, and 26 (11.3%) neonates were unclassified (Table 1). In univariate analysis, signs that were significantly more frequent in the infected group than in the not infected group were a capillary refill time >3 s, hypotonia or lethargy, gray and/or pale skin complexion, fever, and tachycardia (Figure 2a). In multivariate analysis, capillary refill time >3 s was the only sign that was significantly associated with an infection (adjusted OR: 4.02, 95% CI [1.15–15.18], *p*-value 0.029). This sign was present in only 10/51 patients of the infected group, so its sensitivity was 20%. A model combining tachycardia, capillary refill time >3 s, and hypotonia or lethargy showed a partial AUC of 0.517 (95% CI [0.502–0.551]) for the diagnosis of infection (Figure 2b).

The median CRP values were significantly higher in infected patients (13.5 mg/L, range: 0–207) than in not infected patients (1 mg/L, range 0–30, *p*-value 0.001).

The prescription of antibiotics, which was at the discretion of the physician on the basis of medical history of the patient, clinical characteristics and CRP level, concerned half (49.1%) of the neonates involved in the study, including all subjects (100%) classified as infected and 27% of those classified as not infected. Vancomycin was the most prescribed drug (42.6% of the total population), followed by amikacin (34.8%) and cefotaxime (17.8%; Table 1).

Blood culture was positive in 43/51 (84.3%) patients classified as infected, and *Staphylococcus* spp. represented 88.4% (38/43) of identified pathogens. Among the 8 patients with a sterile blood culture classified as infected, pathogens were detected in the tracheal suctioning culture in 4 patients and 2 had a positive blood culture (for either *Pseudomonas aeruginosa* or *Staphylococcus epidermidis*) the day following their inclusion. In the not infected group, 3/153 (2%) patients had a positive blood culture; for these 3 patients, coagulase negative staphylococci were identified.

### 3.3. Biomarkers’ Characteristics

The distribution of concentration of each biomarker is presented for the three groups of patients in Figure 3a and Appendix A. Concerning IL-27, due to the high proportion of missing data (94/230) related to the serum volume requirement of 100 µL, we decided to exclude it from the performance calculation. Considering patients classified as infected and not infected, the AUC was calculated for each biomarker alone (Figure 3b). IL-6, IL-10, and NGAL had the best AUC (>0.8 for all). In line with the clinical context and the need to identify a biomarker useful to rule out the presence of an infection in symptomatic neonates, partial AUC focusing on a high sensitivity was then calculated to evaluate the performance of each biomarker and of all combinations of two to four biomarkers (Appendix A). No added value was obtained when combining four rather than three biomarkers; combinations of more than four biomarkers were therefore not tested (Appendix A). Focusing on partial AUC, the best performance was found for IL-6, IL-10, and NGAL alone, as well as for the combinations PCT/IL-10, PTX3/NGAL, and PTX3/NGAL/gelsolin (Figure 3c). In comparison, the performance of CRP to distinguish infected from not infected patients in the present cohort was lower (AUC 0.765 [95% CI: 0.673, 0.858]), as was the performance of the clinical model (AUC 0.635 [95% CI: 0.549, 0.722]) (Figure 2b and Figure 3b). Of note, these performances were overestimated given that CRP and clinical variables were part of the parameters used by the adjudicators to assign the infection status of patients.

As illustrated in the heatmap (Figure 4), unsupervised analysis revealed a cluster characterized by high plasmatic levels of IL-10, IL-6, and NGAL that was mainly composed of infected (73%) or unclassified (23%) neonates. This indicates that the biomarker profile of patients in the unclassified group was close to the one of the infected group (Figure 3a).

### 3.4. Application of the Best Models to the Cohort

We assessed the reclassification of patients using the identified biomarkers alone and in combination. Using the 6 models with highest partial AUC (Figure 3c), 5/51 (9%) patients of the infected group were reclassified as not infected; this was consistent with the sensitivity of each model that was preset to about 90%. The 5 patients reclassified as not infected varied depending on the model. The 6 models were able to identify up to 64.3% (27/42) of neonates of the not infected group who had received unjustified antibiotics as not infected (Table 2).

## 4. Discussion

In the present study, we tested biomarkers alone or in combination and identified a subset of biomarkers with a high performance to diagnose non-infected neonates among symptomatic patients; the proportion of patients that could have avoided unjustified antibiotic exposure through the implementation of these biomarkers was estimated at up to two-thirds.

Among the six models that had the best performance, three were combinations of biomarkers (PCT/IL-10, PTX3/NGAL, and PTX3/NGAL/gelsolin). As far as we know, this is the first time these combinations have been tested in a neonatal population with suspected LOS. The use of combinations of biomarkers seemed an interesting idea since such combinations benefit from the performance of each biomarker that could have, individually, different advantages and limits. However, the combinations tested herein did not show significantly better performance than biomarkers alone, and we therefore focused the rest of the discussion on biomarkers used alone. IL-6, IL-10, and NGAL showed the best performance. In contrast to IL-10 and NGAL, and despite high AUC and sensitivity, IL-6 surprisingly failed to correctly identify the patients who received unjustified antibiotics as not infected. This is likely due to the close relationship between IL-6 and CRP, since the former is a cytokine of the early immune response that directly stimulates the hepatic production of CRP [24]. Thus, we suggest that the contribution of IL-6 in the reclassification of these patients was moderate because the decision to treat or not treat patients was made by the clinicians on the basis of the CRP value, which increased in parallel to that of IL-6. In contrast, IL-10 seems promising since it could have avoided unjustified antibiotics for two-thirds of patients. This result is consistent with that reported in a previous study exploring the performance of IL-10 for the diagnosis of LOS in a population of full-term neonates [25]. The reason for IL-10′s good performance is likely related to the immune response during the neonatal period, notably in preterm infants, which is polarized towards an anti-inflammatory response (T helper 2 lymphocytes) involving an increased production of cytokines such as IL-10 [26]. NGAL is another biomarker that showed a good performance to identify not infected neonates. NGAL is a protein produced by neutrophils that inhibits bacterial growth by blocking the access of bacteria to iron, and its production is regulated by stimuli different from the ones involved in cytokine production [27]. NGAL has been proposed as a promising early biomarker of invasive neonatal sepsis in a previous study including both term and preterm infants [28]. However, it can be influenced by other neonatal conditions, including respiratory distress and acute kidney injury (AKI) [28,29]. More studies are needed to thoroughly investigate the performance of this biomarker in patients suffering from AKI and to evaluate whether a different threshold value for plasmatic NGAL concentration can be proposed to differentiate AKI from LOS.

We chose to evaluate biomarkers that have already been documented as associated with neonatal infection, but the originality of our study lies in the methodology used herein. First, some studies compared biomarker levels in infected versus healthy neonates [12,15,16,21,30]. However, we consider that the comparison to healthy neonates is not relevant in clinical practice since the real difficulty is to differentiate infected from not infected neonates among those with clinical symptoms. Second, studies about biomarkers are either frequently descriptive about mean and distribution of biomarker levels in a specific population or focused on the overall performance of the biomarker via the measurement of AUC, specificity, and sensitivity [11,12,13,15,16]. However, in clinical practice, the daily issue is not to confirm LOS but to rule it out at the onset of clinical signs. This is illustrated herein as all infected patients had been properly identified as such by clinicians and had all received antibiotics. In this context, we decided to use an original approach; we determined the best partial AUC considering a minimal sensitivity of 0.898, which seems acceptable from a clinician’s point of view to avoid missing the diagnosis of LOS. This innovative approach explains why the threshold value for the biomarkers in the present study differed from that reported elsewhere; for example, the cut-off for IL-10 in our study was 2.5- to 4.5-fold lower than that proposed in previous studies [25,31].

Another point of note is that the study of 11 biomarkers was made possible by the use of the ELLA Automated Immunoassay System, which requires only 25 µL of serum for the quantification of four proteins [32]. Such a low volume of blood is a prerequisite in the specific population of neonates and very low birth weight infants to avoid blood depletion. To the best of our knowledge, this is the first time this method was used in neonates, thus opening new prospects for future research. However, this technique is not applicable to clinical routine. The next step, which is currently underway, is to develop a rapid point of care test. This is mandatory for the implementation of these biomarkers in a clinical decision rule because a quick result is essential in impacting the decision to prescribe or not prescribe antibiotics, as described in a previous study [33]. When this first step is completed, the second step will be to evaluate whether having the biomarker value in neonates suspected of LOS will decrease unjustified antibiotic prescription without missing LOS. The impact on microbiota and on emergent multidrug resistant bacteria in NICU settings will also be an essential outcome to evaluate in future validation studies.

The present study does have some limitations. First, due to the use of CRP by the adjudication committee to classify patients, the performance of CRP should be taken with caution and only be considered as indicative. This also precluded the comparison of our results to those of previous studies evaluating CRP’s performance for LOS diagnosis. However, despite being the most used biomarker for LOS diagnosis in current practice, several studies deplored the poor performance of CRP. Thus, the second limitation is the heterogeneity of the included patients, notably regarding their gestational age and weight at birth, their postnatal age, or their need for surgery. However, the aim was to include all neonates suspected of LOS in order to extrapolate the results to the whole population of hospitalized neonates without restriction. Our results cannot be applied to EOS due to the criteria of age >7 days for inclusion. Third, although published data suggested that it could be promising for the diagnosis of LOS [14], it was not possible to evaluate the performance of IL-27 due to the blood volume required for the test. This cytokine is not currently measurable using ELLA but could be in the future. Finally, despite including a large set of neonates, our study did not include an independent validation cohort; further validation studies are necessary to confirm the candidate biomarkers identified in this exploratory study.

## 5. Conclusions

The present study suggests that the diagnosis of LOS in neonates could be improved by the use of new biomarkers. The next step will be to validate these results in an independent cohort and to evaluate if including these biomarkers in a clinical decision rule could have a positive impact on the adequate prescription of antibiotics in hospitalized neonates.

## 6. Patents

The work reported in this manuscript has been subjected to the two patent applications no. FR2203313 and no. FR2203314.

## Figures and Tables

**Figure 1 biomedicines-11-01703-f001:**
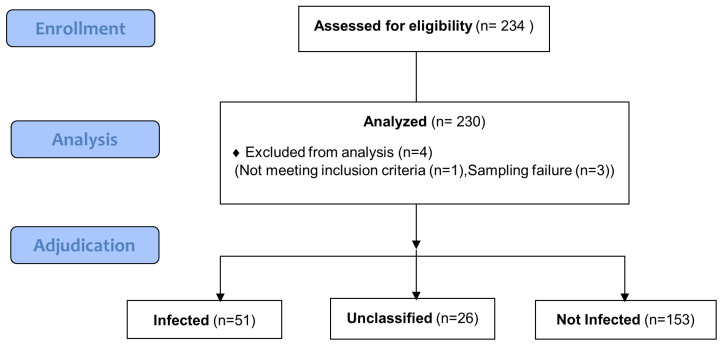
Study flow diagram. Out of 234 enrolled hospitalized neonates, 1 was ineligible due to invalid inclusion criteria and 3 due to sampling failure. Out of the 230 analyzed patients, the adjudication committee classified patients as infected (n = 51), not infected (n = 153), and 26 subjects were unclassified.

**Figure 2 biomedicines-11-01703-f002:**
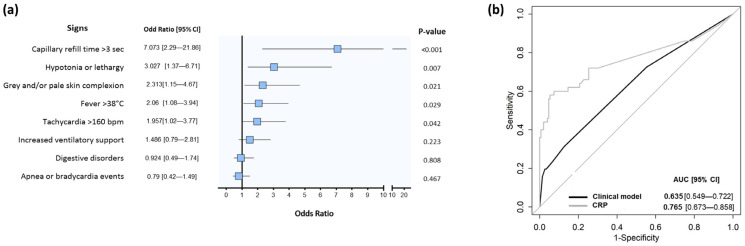
Performance of clinical signs and CRP to discriminate infected and not infected neonates. Forest plot of odds ratio [95% CI] relative to each clinical sign for infection diagnosis is depicted (**a**). The squares represent odds ratio and bars indicate the 95% confidence interval (CI). ROC curve of both CRP and clinical model and best performing model combining tachycardia, capillary refill time >3 s, and hypotonia/lethargy are shown (**b**). AUC [95% CI] is calculated for the diagnosis of confirmed infection. Of note, these ROC curves and AUC are presented only as indicative since these parameters were used by the adjudication committee to classify patients and are thus partially biased.

**Figure 3 biomedicines-11-01703-f003:**
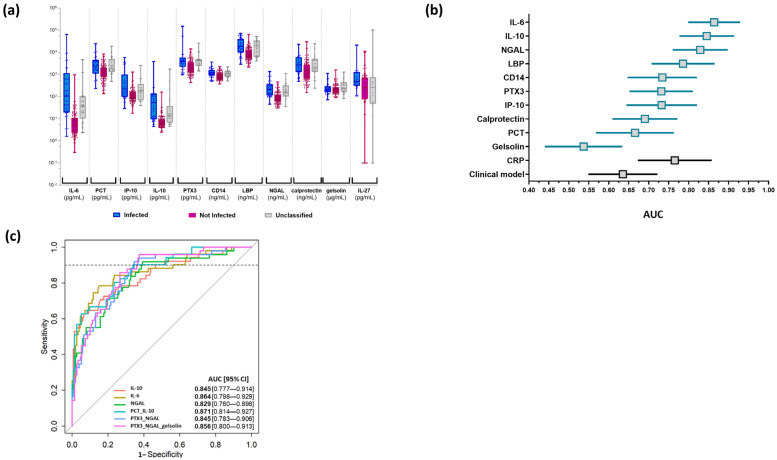
Description and performance of biomarkers to discriminate infected and not infected neonates. (**a**) The concentration of the eleven investigated biomarkers was measured as described in Materials and Methods. Distributions of the biomarker levels in the indicated groups are shown as box and whisker plots. Respective medians (horizontal line inside the box) and interquartile range (upper and lower horizontal lines of the box) values are shown in Appendix A. Each dot corresponds to one subject. (**b**) Forest plots depicting AUC [95% CI] relative to each biomarker for infection diagnosis in comparison to that of CRP and the clinical model. The squares represent AUC and bars indicate the 95% CI. AUC of CRP and clinical models are presented in a different color (black) because performances of these parameters were biased by the fact that they were used by the adjudication committee to classify patients. (**c**) ROC curves are shown for the best performing biomarker alone or in combination for infection diagnosis. The dashed line represents a sensitivity of 0.9, established to calculate partial AUC [95% CI].

**Figure 4 biomedicines-11-01703-f004:**
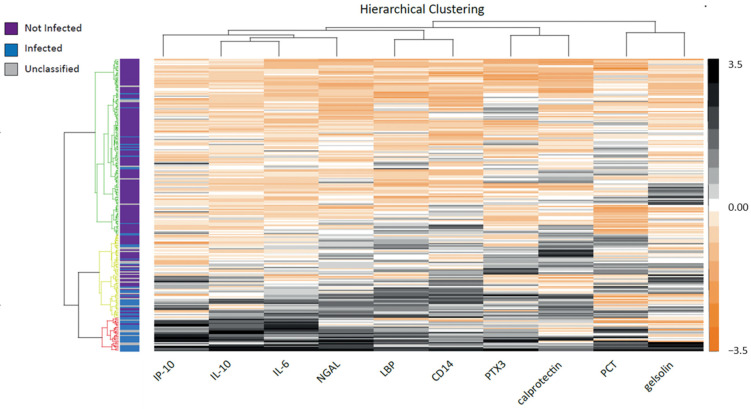
Unsupervised analysis of biomarkers’ quantification in neonates with suspected infection. The heatmap depicts biomarker expression profiles in patients with infected (blue bars), not infected (purple bars), or unclassified status (gray bars) from unsupervised analysis (Euclidean distances matrix with Ward’s methods) generated by scaling and centering log10-transformed normalized biomarker concentrations. Biomarker intensities are displayed as colors ranging from orange to black, as shown in the key. Biomarker clustering is indicated by dendrogram trees on the top and on the left side of the heatmap, respectively. Clustering allows to discriminate 3 distinct clusters (red, green, and yellow dendrogram trees on the left side of the heatmap) composed of various proportions of patients with infected, not infected, or unclassified status. The biomarker profile of patients in the unclassified group was close to the one of the infected group.

**Table 1 biomedicines-11-01703-t001:** Patients’ characteristics.

	All Patients (n = 230)	Infected (n = 51)	Not Infected (n = 153)	Unclassified (n = 26)	*p* Value	Adjusted *p* Value ^1^
**Demographic characteristics, No.** (**%**)						
Sex, male	137 (59.6)	39 (76.5)	86 (56.2)	12 (46.2)	0.011	0.231
Gestational age (in weeks), Median (range)	27 (23–41)	27 (24–41)	28 (23–41)	26.5 (24–38)	0.610	1.000
Birth weight (g), Median (range)	940 (450–4660)	960 (530–3400)	930 (450–4660)	902.5 (490–3430)	0.692	1.000
Birth weight < 1500 g	184 (80.0)	39 (76.5)	126 (82.4)	19 (73.1)	0.393	1.000
Apgar Score at 5 min, Median (range)	8 (1–10)	9 (1–10)	8 (1–10)	8 (4–10)	0.850	1.000
Small for gestational Age	67 (29.1)	10 (19.6)	52 (34.0)	5 (19.2)	0.074	1.000
C-section birth	152 (66.1)	27 (52.9)	109 (71.2)	16 (61.5)	0.051	0.816
Histological chorioamnionitis	36 (16.5)	7 (14.6)	26 (17.8)	3 (12.5)	0.816	1.000
Congenital malformations	41 (17.8)	13 (25.5)	25 (16.3)	3 (11.5)	0.260	1.000
Surgery prior to inclusion	35 (15.2)	17 (33.3)	15 (9.8)	3 (11.5)	0.001	0.028
Time from surgery to inclusion(in days), Median (range)	15.0 (4.0–63.0)	16 (6–63)	15 (6–43)	6 (4–52)	0.566	1.000
**Clinical features at inclusion, No.** (**%**)						
Calculated age (in days), Median (range)	14 (7–178)	11 (7–159)	15 (7–178)	14 (7–69)	0.637	1.000
Fever > 38 °C	84/229 (36.7)	25/51(50)	103/153 (67.3)	17/26 (65.4)	0.087	1.000
Tachycardia > 160 bpm	124/230 (53.9)	33/51 (64.7)	74/153(48.4)	17/26 (65.4)	0.065	0.975
Capillary refill time > 3 s	18/226 (8.0)	10/51 (19.6)	5/150 (3.3)	3/25 (12.0)	0.001	0.028
Grey and/or pale skin complexion	56/227 (24.7)	18/51 (35.3)	29/152 (19.1)	9/24(37.5)	0.020	0.360
Apnea or bradycardia events	111/229 (48.5)	23/51 (45.1)	78/153 (51.0)	10/25 (40.0)	0.534	1.000
Digestive disorders ^2^	120/230 (52.2)	26/51 (51.0)	81/153 (52.9)	13/26 (50.0)	0.938	1.000
Hypotonia or lethargy	38/229 (16.6)	14/51 (27.5)	17/153 (11.1)	7/25 (28.0)	0.006	0.132
Increased ventilatory support and/or increased FiO2	107/230 (46.5)	26/51 (51.0)	63/153 (41.2)	18/26 (69.2)	0.022	0.374
Cutaneous rash	5/230 (2.2)	1/51 (2.0)	3/153(2.0)	1/26 (3.8)	0.782	1.000
Presence of a central venous catheter	146/229 (63.8)	44/50 (88.0)	86/153 (56.2)	16/26 (61.5)	0.001	0.028
**Antibiotics at 48 h, No.** (**%**)						
No	117 (50.9)	0 (0)	111 (72.5)	6 (23.1)	0.001	0.028
Yes	113 (49.1)	51 (100)	42 (27.5)	20 (76.9)	NA	NA
Vancomycin	98 (42.6)	48 (94.1)	36 (23.5)	14 (53.8)	NA	NA
Amikacin	80 (34.8)	35 (68.6)	32 (20.9)	13 (50.0)	NA	NA
Cefotaxime	41 (17.8)	20 (39.2)	13 (8.5)	8 (30.8)	NA	NA
Other betalactams	18 (7.8)	8 (15.7)	5 (3.3)	5 (19.2)	NA	NA
Metronidazole	2 (0.9)	1 (2.0)	1 (0.7)	0 (0)	NA	NA
Other	20 (8.7)	9 (17.6)	6 (3.9)	5 (19.2)	NA	NA
Duration of exposure (days), Median (range)	3 (1–26)	10 (2–21)	2 (2–26)	3 (2–21)	NA	NA
Antibiotic exposure > 2 days	64 (66)	48 (94)	7 (17)	9 (45)	NA	NA
**Laboratory values**						
C-reactive protein, mg/L, (n = 187) Median (range)	1.0 (0.0–207.0)	13.5 (0–207)	1 (0–30.0)	5.6 (0–165.9)	0.001	0.028
White blood cell count, G/L, (n = 133) Median (range)	13.3 (2.30–40.12)	14.48 (2.30–40.12)	12.65 (2.94–38.05)	16.67 (5.67–33.3)	0.205	1.000
Neutrophils, G/L, (n = 106) Median (range)	5.13 (0.93–22.45)	6.50 (0.95–22.07)	4.61 (0.93–22.45)	4.70 (1.01–21.98)	0.015	0.285
Lymphocytes, G/L, (n = 106) Median (range)	5.06 (0.77–14.90)	3.81 (0.77–6.73)	5.39 (1.14–14.90)	5.01 (1.17–7.93)	0.012	0.240
**Blood cultures No./n.** (**%**)						
Not done	2/230 (0.9)	0/51 (0)	2/153 (1.3)	0/26 (0)	0.001	0.028
Sterile	180/230 (78.0)	8/51 (15.7)	148/153 (96.7)	24/26 (92.3)	NA	NA
Positive	48/230 (20.9)	43/51(84.3)	3/153 (2)	2/26 (7.7)	NA	NA
*Staphylococcus aureus* (n = 228)	8/228(3.5)	8/51 (15.7)	0/151 (0)	0/26 (0)	NA	NA
Coagulase-negative staphylococci (n = 228)	35/228(15.4)	30/51 (58.8)	3/151 (2.0)	2/26 (7.7)	NA	NA
Gram-negative bacilli (n = 228)	3/228(1.3)	3/51 (5.9)	0/151 (0)	0/26 (0)	NA	NA
Other Gram-positive organisms (n = 228)	2/228(0.9)	2/51 (3.9)	0/151 (0)	0/26 (0)	NA	NA
*Candida albicans* (n = 228)	1/228(0.4)	1/51 (2.0)	0/151 (0)	0/26 (0)	NA	NA

^1^ Holm’s adjusted *p*-value. ^2^ Abdominal bloating or rectal bleeding. NA: Not appropriate.

**Table 2 biomedicines-11-01703-t002:** Reclassification by selected models of patients treated by antibiotics.

Selected Models	Patients Who Received Antibiotics	
	Patients from the Infected Group, Reclassified as Not Infected Using Biomarker Models (n/N, %)	Patients from the Not Infected Group, Also Classified as Not Infected Using Biomarker Models (n/N, %)
IL-6	5/51 (9.8%)	10/42 (23.8%)
IL-10	5/51 (9.8%)	26/42 (61.9%)
NGAL	5/49 (10.2%)	25/42 (59.5%)
PCT/IL-10	5/51 (9.8%)	26/42 (61.9%)
PTX3/NGAL	5/49 (10.2%)	27/42 (64.3%)
PTX3/NGAL/gelsolin	5/49 (10.2%)	23/41 (56.1%)
IL-6	5/51 (9.8%)	10/42 (23.8%)

Data represent the proportion of patients reclassified using the best selected models among patients treated by antibiotics (51 and 42 neonates classified by adjudication committee as patients with infected and not infected status, respectively). There is missing data for 3 biomarkers’ detection.

## Data Availability

Study protocol, blank informed consent form, blank Case Report Form, and deidentified individual participant dataset are available upon request. Requests for data should be made to the corresponding author. The datasets generated and analyzed during the current study are available from the corresponding author on reasonable request.

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
