# Peer review of "Performance of 11 Host Biomarkers Alone or in Combination in the Diagnosis of Late-Onset Sepsis in Hospitalized Neonates: The Prospective EMERAUDE Study"

_biomedicines, 2023, doi:10.3390/biomedicines11061703_

Round 1

Reviewer 1 Report

This is a well-done study evaluating the use of several different biomarkers in the diagnosis of LOS. However, some questions remain:

What was the rationale for the selection of investigated biomarkers?
What was the rationale for the selection and omission of clinical signs of infection? Hypothermia and Central-peripheral temperature difference, for example, were not considered.

The decision process by the adjudication committee could be described in more detail.

Teh biomarker combinations were selected acording to the existing data, it is unknown whether these combinations perform as well in other patient groups, so interpretation must be very cautious.
A specific statement that the results do not apply to EOS may also be helpful

Line 102: There were two NICUs in the study. What about the other NICU?
Line 116 "antibiotic" should not be plural here.

Fig 4 ist difficult to comprehend for readers not familiar with the method.

Good English, one typo found

Reviewer 2 Report

The article, titled: Performance of 11 host biomarkers alone or in combination in the diagnosis of late-onset sepsis in hospitalized neonates: the prospective EMERAUDE study touches upon an important issue. The neonate’s LOS is a difficult diagnosis and the avoidance of unnecessary antibiotic treatment is an important aspect, but the non-introduction of antibiotic treatment might be very dangerous as well.  The biomarker model developed and introduced by the authors might serve as an important approach to selecting between infected and noninfected neonates. The study is extensive and well-designed, appropriately documented, and the results are adequately explained.  
